# Prevalence of Osteonecrosis of the Jaw Following Tooth Extraction in Patients with Osteoporosis: A Systematic Review and Meta-Analysis

**DOI:** 10.3390/jcm14175988

**Published:** 2025-08-25

**Authors:** Evangelos Kostares, Georgia Kostare, Michael Kostares, Fani Pitsigavdaki, Christos Perisanidis, Maria Kantzanou

**Affiliations:** 1Department of Microbiology, Medical School, National and Kapodistrian University of Athens, 115 27 Athens, Greece; gkostare@med.uoa.gr (G.K.); mkatzan@med.uoa.gr (M.K.); 2Department of Anatomy, Medical School, National and Kapodistrian University of Athens, 115 27 Athens, Greece; michaliskost@med.uoa.gr; 3Department of Oral and Maxillofacial Surgery, Dental School, National and Kapodistrian University of Athens, 115 27 Athens, Greece; fpitsi@dent.uoa.gr (F.P.); cperis@dent.uoa.gr (C.P.)

**Keywords:** medication-related osteonecrosis of the jaw, MRONJ, osteoporosis, tooth extraction, prevalence, meta-analysis

## Abstract

**Background/Objectives**: Medication-related osteonecrosis of the jaw (MRONJ) is a rare but serious adverse effect associated with antiresorptive therapy, particularly following tooth extraction in osteoporotic patients. This study aimed to estimate the prevalence of MRONJ after tooth extraction in individuals with osteoporosis through a systematic review and meta-analysis. **Methods**: A comprehensive search was conducted across three major databases (Medline, Scopus, Web of Science) up to January 2025, including observational studies reporting MRONJ following extractions in osteoporotic patients treated with antiresorptives. Pooled prevalence rates were calculated using a random-effects model, and study quality was assessed. Influence analyses as well as meta-regression were also performed. **Results**: Twenty-four studies were included, comprising 3784 patients and 5426 extractions. The pooled prevalence of MRONJ was 1.7% (95% confidence intervals (CI): 0.8–3.0%), with considerable heterogeneity (I^2^ = 77%). When considering only cases of bisphosphonate-related osteonecrosis of the jaw (BRONJ) following tooth extraction in patients with osteoporosis, the estimated prevalence was 0.7% (95% CI: 0.1–1.8%), with substantial heterogeneity (I^2^ = 61%). No individual study was identified as overly influential. Meta-regression revealed no significant associations between MRONJ prevalence and variables such as publication year, gender proportion, or extraction-to-patient ratio. **Conclusions**: This meta-analysis underscores the importance of vigilance in managing osteoporotic patients undergoing extractions, emphasizing the need for consistent diagnostic criteria and preventive protocols to mitigate MRONJ risk.

## 1. Introduction

Osteoporosis is a relatively common metabolic bone disease characterized by low bone mineral density and compromised bone microarchitecture, increasing the risk of fragility fractures from minimal trauma [1]. A recent systematic review and meta-analysis estimated the global prevalence of osteoporosis at 19.7% and osteopenia at 40.4%, revealing considerable variation across countries and regions [2]. It is classified into primary and secondary forms. Primary osteoporosis typically results from the natural aging process combined with declining levels of sex hormones, leading to the progressive deterioration of bone structure. Secondary osteoporosis, more common in men, arises from other medical conditions or it can result from the effects of medications and nutritional deficiencies. Common contributors include medications like glucocorticoids and anti-epileptics, with other agents such as chemotherapy drugs, proton pump inhibitors, and thiazolidinediones also suspected. Various diseases, including hyperparathyroidism, chronic renal failure, hyperthyroidism, and Cushing’s syndrome, as well as prolonged immobilization, can lead to secondary osteoporosis. Additionally, factors such as anorexia, malabsorption, and secondary amenorrhea lasting more than a year due to low body weight, excessive exercise, or hormonal therapies can accelerate bone loss [1].

First-line treatment for osteoporosis includes antiresorptive agents such as bisphosphonates (e.g., risedronate, alendronate, zoledronic acid) and denosumab, both of which significantly reduce the risk of vertebral and non-vertebral fractures. Bisphosphonates are also recommended for men with osteoporosis. These agents work by inhibiting osteoclast-mediated bone resorption by attaching to hydroxyapatite binding sites on the bone, particularly in areas with active resorption, thereby increasing bone mineral density and reducing fracture risk. Bisphosphonates can be administered orally or intravenously and are also used in managing other skeletal conditions, including Paget’s disease of bone and osteogenesis imperfecta [1,3,4]. Denosumab, a total human IgG2 monoclonal antibody that binds to the receptor activator of NF kappa B ligand (RANKL) and competitively inhibits its binding to the RANK, also inhibits osteoclast activity and is administered subcutaneously every six months for osteoporosis. It provides similar fracture risk reduction and is especially beneficial in patients with renal impairment, where bisphosphonates may be contraindicated. Unlike bisphosphonates, denosumab does not bind to bone, and its effects diminish relatively quickly after cessation [1,5]. Additionally, romosozumab, a humanized monoclonal antibody, enhances bone formation and reduces resorption by inhibiting sclerostin through the Wnt signaling pathway, offering another therapeutic option for postmenopausal women at high risk of fracture. Another anabolic therapy is abaloparatide, a synthetic analog of parathyroid hormone–related protein (PTHrP) that stimulates bone formation and has been shown to significantly reduce vertebral and nonvertebral fracture risk; it is approved for postmenopausal women at high risk of fracture and is generally limited to a maximum of two years of use [1,3,6].

A specific adverse event associated with antiresorptive drugs (ARDs) is termed medication-related osteonecrosis of the jaw (MRONJ). MRONJ, as defined by the American Association of Oral and Maxillofacial Surgeons (AAOMS) in its 2022 update, is characterized by exposed bone or bone that can be probed through an intraoral or extraoral fistula in the maxillofacial region persisting for more than eight weeks, in a patient with current or previous treatment with antiresorptive therapy alone or in combination with immune modulators or antiangiogenic agents, and with no history of radiation therapy to the jaws or metastatic disease to the jaws [3]. According to the Italian position paper by SIPMO–SICMF (2024), MRONJ is diagnosed when all three of the following criteria are present: (1) current or previous treatment with bone-modifying agents and/or antiangiogenic agents, (2) clinical and radiological findings of progressive bone destruction, and (3) no history of radiation therapy to the jaws or primary/metastatic malignancy involving the jaws [7].

MRONJ is a rare but serious condition, with prevalence rates varying from less than 0.05% in osteoporotic patients to up to 5% in cancer patients. Its pathophysiology is multifactorial, involving suppression of bone remodeling, infection or inflammation, inhibition of angiogenesis, immune dysfunction, and potential genetic predisposition. Mechanistically, MRONJ is thought to result primarily from potent inhibition of osteoclastic bone resorption by antiresorptive agents such as bisphosphonates and denosumab, leading to impaired removal of necrotic bone and defective healing following dental disease or trauma. This promotes local accumulation of nonviable osteocytes, sustained pro-inflammatory immune responses (including Th17 cell activation and M1 macrophage polarization), and disruption of mucosal integrity, often compounded by microbial colonization of exposed bone [6]. Preventive strategies are crucial and should be implemented both before and during antiresorptive therapy. Before treatment initiation, comprehensive dental assessment (including radiographic evaluation), the elimination of oral infections, extraction of non-restorable teeth, and the adjustment of ill-fitting dentures are recommended. During therapy, patients should maintain excellent oral hygiene, attend regular dental check-ups (every 6 months), promptly report oral symptoms, and avoid invasive dental procedures unless absolutely necessary, with expert consultation sought when such interventions are unavoidable. These measures, combined with effective patient education on the signs and symptoms of MRONJ, have been shown to substantially reduce the risk of developing the condition [1,8,9].

Further insights were provided by a meta-analysis conducted by Srivastava et al. [9], which compared the prevalence of MRONJ in patients receiving sequential versus single-agent antiresorptive therapies. Among those treated sequentially with pamidronate followed by zoledronate (9 studies), the pooled MRONJ prevalence was 19% (95% CI: 10–27%; I^2^ = 91.28%, *p* < 0.01). In contrast, patients treated with pamidronate alone had a markedly lower prevalence of 1% (95% CI: 0–3%; I^2^ = 46.44%, *p* = 0.06), and those receiving only zoledronate showed a prevalence of 7% (95% CI: 3–10%; I^2^ = 83.40%, *p* < 0.01). Notably, sequential bisphosphonate-denosumab therapy was associated with a pooled MRONJ prevalence of 13% (95% CI: 3–22%; I^2^ = 91.28%, *p* < 0.01), which exceeded the prevalence observed in bisphosphonate-only (5%; 95% CI: 0–9%) and denosumab-only groups (4%; 95% CI: 3–5%).

The present systematic review and meta-analysis aim to estimate the prevalence of MRONJ in osteoporotic patients following dental extractions. Additionally, we investigate potential sources of heterogeneity to better elucidate the factors contributing to MRONJ risk in this population.

## 2. Materials and Methods

### 2.1. Search Strategy

A systematic literature search was independently conducted by two reviewers in Medline/PubMed Central (PMC) (via PubMed), Scopus, and Web of Science databases, covering publications up to 30 January 2025. The study adhered to the structure and reporting standards outlined in the Preferred Reporting Items for Systematic Reviews and Meta-Analyses (PRISMA) guidelines. The PRISMA checklist is available in the Appendix A (Appendix A). The search algorithm included keywords such as bisphosphonate, denosumab, medicat*, antiresorptive, osteonecrosis, MRONJ, extrac*, mandib*, maxill*, tooth, and osteoporo*. The complete search strategy for each database is provided in the Appendix A (Appendix A).

Two independent researchers thoroughly examined the publications identified in the initial search. First, duplicates were removed. The titles and abstracts of the remaining articles were then screened, and publications that did not meet the predetermined inclusion criteria were excluded. The full texts of the remaining articles were obtained and carefully assessed. During this process, the reference lists of eligible articles were reviewed to identify any potentially missed studies. Zotero reference management software was used to support this process. Any discrepancies were resolved through team consensus.

### 2.2. Criteria for Study Selection and Data Extraction

To ensure methodological rigor, clarity, and precision in our systematic review and proportional meta-analysis, we formulated the inclusion criteria based on the Population, Exposure, Comparison, Outcomes, and Study Types (PECOS) framework, which provides a structured and transparent approach to study selection (Table 1).

### 2.3. Data Extraction

The following information was collected from each study: primary author’s name, publication year, study design, continent and country of origin, study duration, total number of patients with osteoporosis, extraction-to-patient ratio, gender distribution, mean age, and the number of patients who developed medication-related osteonecrosis of the jaw following tooth extraction and antiresorptive therapy.

### 2.4. Quality Assessment

Two independent researchers evaluated each study using the Newcastle-Ottawa Scale (NOS)—the original version for cohort studies and a modified version for cross-sectional studies. The assessment examined three main domains: selection of study groups, group comparability, and ascertainment of exposure or outcome. A star-based scoring system was applied, classifying studies as high quality (7–9 stars), moderate quality (4–6 stars), or low quality (0–3 stars). The aim was to detect methodological or survey-related issues that could affect internal validity.

### 2.5. Statistical Analysis

Statistical analysis was performed in RStudio (v2022.12.0+353) using the metafor package. Pooled prevalence with 95% CIs was estimated via the DerSimonian–Laird random-effects model, applying the Freeman–Tukey double arcsine transformation. Heterogeneity was evaluated through forest plot inspection, Cochran’s Q test (with *p* value), and Higgins’ I^2^ statistic (with 95% CI), classified as not important (0–40%), moderate (30–60%), substantial (50–90%), or considerable (75–100%). Identifying influential outlying effect sizes involved screening for externally studentized residuals and leave-one-out diagnostics. Subgroup analysis for variables such as continent of origin was not conducted due to the difficulty of categorizing studies conducted in more than one continent. Meta-regression analysis was conducted when at least ten studies were available for a given continuous or categorical variable. However, variables such as mean age and the route of bisphosphonate administration (oral vs. intravenous) were excluded due to insufficient data (fewer than ten studies). Statistical significance was set at *p*  =  0.05 (two-tailed). Publication bias tests such as Egger’s, Begg’s, and funnel plots were not used because, in proportion meta-analyses, definitions of positive results are unclear; a qualitative assessment was applied instead.

## 3. Results

### 3.1. Results and Characteristics of the Included Studies

Of 1851 articles identified in the databases, after removing duplicate articles, excluding studies based on title and abstract screening, and further excluding articles that did not meet our inclusion criteria upon full-text assessment, a total of 24 studies were included in the quantitative analysis. The PRISMA flow chart illustrating this selection process is presented in Figure 1. Table 2 summarizes their descriptive characteristics. The included studies were published between 2008 and 2024 and conducted between 2006 and 2024. The total number of patients with osteoporosis receiving antiresorptive medication who underwent tooth extraction was 3784, accounting for a total of 5426 extractions. Information regarding gender was available in 11 studies, with males comprising 10.6% of patients. Mean age was reported in 7 studies, with a median value of 70.5 years. In 13 studies, bisphosphonates were the only antiresorptive medications used. All studies were assessed as being of moderate quality.

### 3.2. Prevalence of MRONJ Following Tooth Extraction in Patients with Osteoporosis

A random-effects model analysis estimated the prevalence of MRONJ following tooth extraction in patients with osteoporosis at 1.7% (95% CI: 0.8–3%), with considerable heterogeneity between studies (I^2^ = 77%, 95% CI: 53–87%, *p* < 0.001) (Figure 2). Influence diagnostics and the forest plot illustrating the results of the leave-one-out analysis are presented in the Appendix A (Appendix A). According to these analyses, no study was identified as influential. When considering only cases of bisphosphonate-related osteonecrosis of the jaw (BRONJ) following tooth extraction in patients with osteoporosis, the estimated prevalence was 0.7% (95% CI: 0.1–1.8%), with substantial heterogeneity (I^2^ = 61%, 95% CI: 14–85%) (Figure 3).

### 3.3. Meta-Regression Analysis

To investigate potential sources of heterogeneity, a meta-regression analysis was conducted evaluating the impact of continuous variables on the prevalence of MRONJ following tooth extraction. Specifically, the year of publication, extraction-to-patient ratio, and proportion of male patients were examined. None of these variables demonstrated a statistically significant association with MRONJ prevalence (Appendix A).

## 4. Discussion

Our study revealed the prevalence of MRONJ following tooth extraction in patients with osteoporosis of 1.7%, with considerable heterogeneity. The confidence interval for I^2^ ranged from 53% to 87%, indicating considerable to substantial heterogeneity. Many factors could contribute to this, such as geographical region, the time period when the studies were conducted, healthcare system differences, lack of clearly defined guidelines for managing these patients, variations in osteoporosis medications and treatment duration, differences in diagnostic criteria, diverse extraction procedures, and varying medical comorbidities. Through meta-regression analysis, we explored the association between continuous variables and the prevalence of MRONJ, including year of publication, extraction-to-patient ratio, and proportion of males; however, no statistically significant associations were found.

A meta-analysis by Lee S.-H. et al. [42] reported that non-cancer patients receiving bisphosphonates for osteoporosis prevention had a significantly increased risk of developing MRONJ, with a pooled odds ratio (OR) of 2.32 (95% CI: 1.38–3.91; I^2^ = 91%) compared with non-users. Nevertheless, the benefits of fracture prevention are generally considered to outweigh this risk. In another systematic review and meta-analysis, Gaudin E. et al. [43] evaluated MRONJ occurrence following dental extraction separately for patients receiving antiresorptive drugs (ARDs) intravenously for oncologic indications and orally for osteoporosis. For oncologic patients treated intravenously (10 studies, *n* = 564), the pooled prevalence of MRONJ was 3.2% (95% CI: 1.7–4.7%; I^2^ = 74.3%, *p* < 0.0001), whereas among osteoporotic patients receiving oral bisphosphonates (8 studies, *n* = 2098), the pooled prevalence was markedly lower at 0.15% (95% CI: 0.0–0.36%; I^2^ = 7.7%).

In our systematic review and meta-analysis, we included 24 studies involving 3784 osteoporotic patients undergoing a total of 5426 dental extractions, yielding a pooled prevalence of MRONJ of 1.7% (95% CI: 0.8–3%) while that of BRONJ was 0.7% (95% CI: 0.1–1.8%). This estimate contrasts notably with the much lower prevalence of 0.15% (95% CI: 0.0–0.36%) reported by Gaudin et al. [43], Several factors could explain this discrepancy, including differences in the number and selection of studies, patient population characteristics, data transformation methods, databases searched, types of antiresorptive medications evaluated, and variations in the surgical extraction techniques used across the included studies.

The treatment of MRONJ, aims to prevent disease progression, manage symptoms, and support the patient’s ongoing medical therapy. Treatment is stage-specific and includes conservative measures such as antimicrobial mouth rinses, antibiotics, and pain control in early stages (Stage 1 and 2). Surgical intervention, including debridement or resection, may be necessary in more advanced cases (Stage 3). Preventive strategies are crucial and focus on optimizing oral health prior to initiating antiresorptive or antiangiogenic therapies, avoiding invasive dental procedures during treatment, and educating patients on maintaining good oral hygiene. A multidisciplinary approach involving dental and medical professionals is emphasized to ensure balanced management of both MRONJ risk and the primary systemic condition [3,43]. In a meta-analysis conducted by dos Santos Ferreira et al. [44], the effectiveness of teriparatide (TPTD) therapy in treating MRONJ was evaluated across 26 studies involving 111 patients. The pooled data indicated that total resolution of MRONJ was achieved in 59.5% of cases, with better outcomes observed when TPTD was used in combination with other therapeutic modalities, such as antibiotic therapy or surgery. Specifically, patients receiving TPTD in conjunction with another therapy were 1.21 times more likely to experience total resolution compared to those treated with TPTD alone (95% CI: 1.04–1.40; *p* = 0.010). Furthermore, patients with stage 1 MRONJ had significantly higher odds of resolution than those with stage 3 (OR: 1.21; 95% CI: 1.02–1.43; *p* = 0.023), supporting the potential utility of TPTD, particularly in early-stage disease.

### Study’s Limitations

Despite employing a random-effects model to account for between-study variability, the considerable heterogeneity observed in this meta-analysis suggests that the pooled prevalence estimates should be approached with caution. The varied outcomes were expected given the nature of these studies [45,46]. Most included studies predated the 2022 AAOMS Position Paper and therefore could not apply its updated diagnostic criteria, staging, or preventive protocols. Studies were conducted across various geographic regions, each with distinct epidemiological profiles and healthcare systems, limiting the generalizability of the findings. The lack of clearly defined guidelines for managing these patients, variations in osteoporosis medication and duration of treatment, differences in diagnostic criteria, discontinuation of therapy, the possible presence of patients with secondary rather than primary osteoporosis, diverse extraction procedures, and varying medical comorbidities among patients all contribute to increased heterogeneity. Furthermore, variables such as mean age, MRONJ stage, elapsed time from extraction to MRONJ diagnosis, the use of antibiotics, the use of platelet-rich fibrin, the type of sutures, and treatment duration were either inconsistently reported, presented in incompatible formats, or available in too few studies to allow a robust meta-regression analysis. Additionally, only observational studies were included. Another significant limitation was the exclusion of non-English language studies and those without full-text availability, potentially introducing selection bias. Moreover, our meta-analysis has not been registered in PROSPERO, which may be a source of bias. Given these considerations, it is important to acknowledge that, like any meta-analysis, our study is subject to potential biases, including selection bias, language bias, search bias, heterogeneity bias, confounding bias, and data extraction bias, all of which may influence the validity and generalizability of our findings.

## 5. Conclusions

This systematic review and meta-analysis provides an updated and comprehensive estimate of the prevalence of MRONJ following tooth extraction in patients with osteoporosis receiving antiresorptive therapy. Our findings suggest a pooled prevalence of 1.7%, a figure higher than previously reported estimates, potentially reflecting variations in study design, diagnostic criteria, treatment regimens, and patient demographics. Although MRONJ remains a relatively uncommon complication, its clinical implications warrant increased awareness among healthcare providers, particularly those involved in the dental management of osteoporotic patients. Early identification, preventive strategies, and coordinated care among dental and medical professionals remain essential. Future studies should aim to reduce heterogeneity by standardizing diagnostic criteria and reporting practices and explore patient- and treatment-specific risk factors through well-designed prospective research.

## Figures and Tables

**Figure 1 jcm-14-05988-f001:**
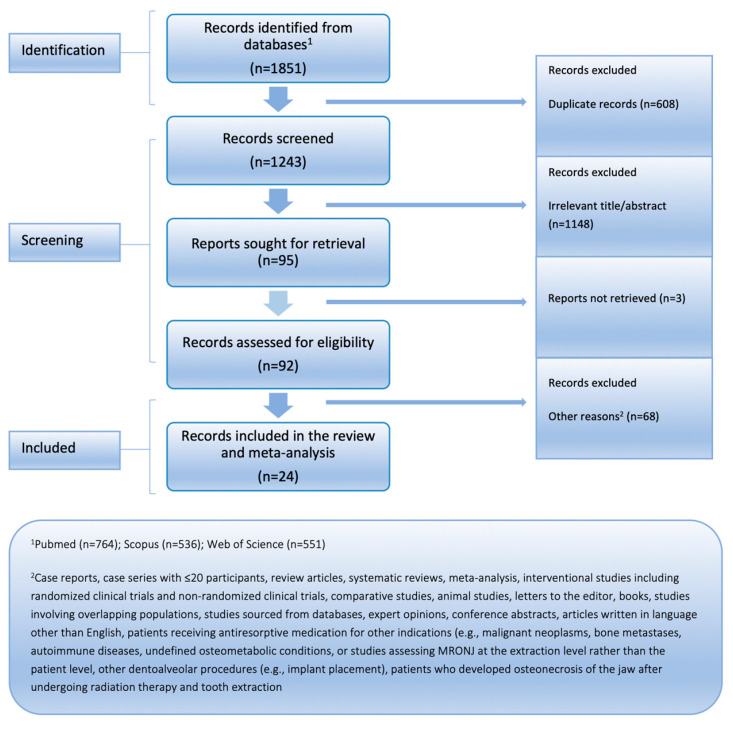
PRISMA flowchart.

**Figure 2 jcm-14-05988-f002:**
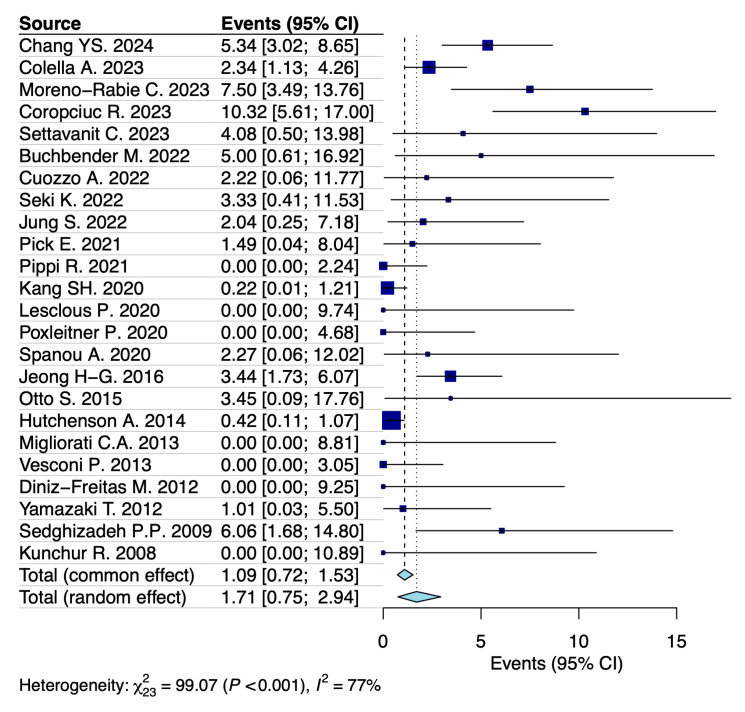
Forest plot illustrating the prevalence of MRONJ following tooth extraction in patients diagnosed with osteoporosis [18,19,20,21,22,23,24,25,26,27,28,29,30,31,32,33,34,35,36,37,38,39,40,41].

**Figure 3 jcm-14-05988-f003:**
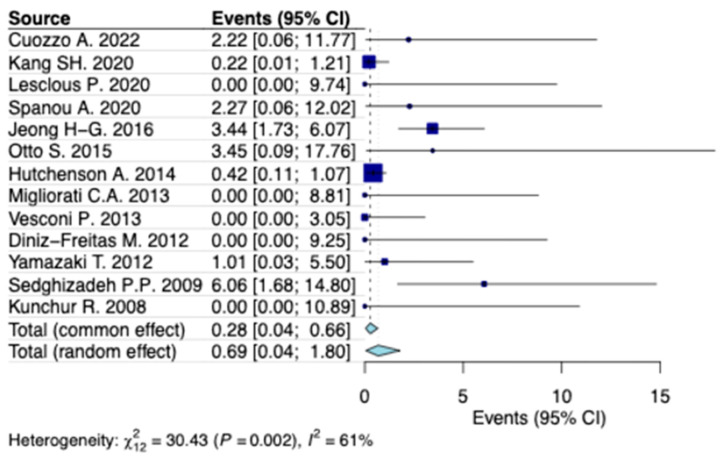
Forest plot illustrating the prevalence of BRONJ following tooth extraction in patients diagnosed with osteoporosis [24,29,30,32,33,34,35,36,37,38,39,40,41].

**Table 1 jcm-14-05988-t001:** Eligibility Criteria.

Eligibility Criteria	Inclusion Criteria	Exclusion Criteria
Population (P)	Patients with osteoporosis receiving antiresorptive medication	Patients receiving antiresorptive medication for other indications (e.g., malignant neoplasms, bone metastases, autoimmune diseases [10], undefined osteometabolic conditions [11], or studies assessing MRONJ at the extraction level rather than the patient level [12])
Exposure (E)	Tooth extractions	Other dentoalveolar procedures (e.g., implant placement [13])
Comparison (C)	As our aim was to quantify prevalence, direct comparisons were not applicable	-
Outcomes (O)	Patients with medication-related osteonecrosis of the jaw following tooth extraction	Patients who developed osteonecrosis of the jaw after undergoing radiation therapy and tooth extraction
Study Types (S)	Observational studies including cohort, case–control, and cross-sectional studies	Case reports, case series with ≤20 participants [14], review articles, systematic reviews, meta-analysis, interventional studies including randomized clinical trials and non-randomized clinical trials [15], comparative studies, animal studies, letters to the editor, books, studies involving overlapping populations [16], studies sourced from databases [17], expert opinions, conference abstracts, articles written in language other than English and studies without full-text availability

**Table 2 jcm-14-05988-t002:** Detailed characteristics of the studies that were included in the evaluation.

First Author	Year of Publication	Study’s Design	Continent of Origin	Country	Study Period	Patients with Osteoporosis	Extractions to Patients ratio	Proportion of Males (%)	Mean Age (Years)	MRONJ	Quality Assessment
Chang YS. [18]	2024	Cohort	Australia	Australia	2020–2024	281	NA	NA	NA	15	Moderate
Colella A. [19]	2023	Cohort	Australia	Australia	2017–2021	427	2.5	17.1	76	10	Moderate
Moreno-Rabie C. [20]	2023	Case–control	Europe	Belgium	2010–2019	120	3	17.5	69	9	Moderate
Coropciuc R. [21]	2023	Cross-sectional	Europe	Belgium	2010–2021	126	NA	19	NA	13	Moderate
Settavanit C. [22]	2023	Cross-sectional	Asia	Thailand	2014–2020	49	NA	NA	NA	2	Moderate
Buchbender M. [23]	2022	Cross-sectional	Europe	Germany	2010–2017	40	NA	NA	NA	2	Moderate
Cuozzo A. [24]	2022	Cohort	Europe	Italy	2019	45	3.5	4.4	67.5	1	Moderate
Seki K. [25]	2022	Cohort	Asia	Japan	2015–2021	60	1.2	6.7	77.5	2	Moderate
Jung S. [26]	2022	Cohort	Asia	Korea	2016–2020	98	1.9	11.2	70.5	2	Moderate
Pick E. [27]	2021	Cohort	Europe	Switzerland	2015–2020	67	NA	NA	NA	1	Moderate
Pippi R. [28]	2021	Case-series	Europe	Italy	2007–2019	163	NA	NA	NA	0	Moderate
Kang SH. [29]	2020	Cohort	Asia	Korea	2008–2017	458	NA	NA	NA	1	Moderate
Lesclous P. [30]	2020	Cohort	Europe	France	NA	36	1	0	NA	0	Moderate
Poxleitner P. [31]	2020	Cohort	Europe	Germany	2017–2019	77	NA	1.3	NA	0	Moderate
Spanou A. [32]	2020	Cohort	Europe	Germany	2008–2012	44	NA	NA	NA	1	Moderate
Jeong H-G. [33]	2016	Cohort	Asia	Korea	NA	320	2	6.9	NA	11	Moderate
Otto S. [34]	2015	Cohort	Europe	Germany	2007–2013	29	2.9	NA	NA	1	Moderate
Hutchenson A. [35]	2014	Cohort	Australia	Australia	2007–2013	950	2.6	33	71	4	Moderate
Migliorati C.A. [36]	2013	Cohort	North America, Europe	USA, Canada, Norway	2007–2011	40	NA	NA	NA	0	Moderate
Vesconi P. [37]	2013	Cross-sectional	Europe	Italy	2006–2010	119	2.6	NA	NA	0	Moderate
Diniz-Freitas M. [38]	2012	Cross-sectional	Europe	Spain	2009–2010	38	NA	0	68.7	0	Moderate
Yamazaki T. [39]	2012	Cohort	Asia	Japan	2006–2009	99	NA	NA	NA	1	Moderate
Sedghizadeh P.P. [40]	2009	Cohort	North America	USA	NA	66	NA	NA	NA	4	Moderate
Kunchur R. [41]	2008	Cross-sectional	Australia	Australia	NA	32	1	NA	NA	0	Moderate

NA: not applicable, MRONJ: medication-related osteonecrosis of the jaw.

## Data Availability

The original contributions presented in this study are included in the article/Appendix A. Further inquiries can be directed to the corresponding author.

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
