# Peer review of "Prevalence of Osteonecrosis of the Jaw Following Tooth Extraction in Patients with Osteoporosis: A Systematic Review and Meta-Analysis"

_jcm, 2025, doi:10.3390/jcm14175988_

Round 1
Reviewer 1 Report
Comments and Suggestions for Authors
Thank you for the possibility to review the paper titled “Prevalence of Osteonecrosis of the Jaw following Tooth Extraction in Patients with Osteoporosis: A Systematic Review and Meta-Analysis” . I have read the paper and have some comments:
Abstract: line 18 – in this population?
Lines 89-105 should be moved to the discussion section
In the introduction section, it should give the reader more information about possible mechanisms of MRONJ , prevention strategy and therapeutic strategies .
In tabl 1. There are some small mistakes regarding study period at 4-th position
Regarding the included paper there are only 5 more or less up to date articles in the way that they were published after 2022 AAOMS Position Paper on Medication-Related Osteonecrosis, but when we compare the observation period of those paper only one is after 2022. This have few implication for the clinical importance of this paper . Some new drugs are not mentioned or included in the paper like for example “Abaloparatide” for other drugs already reported risk level and adverse effect are not included in this paper for example “Romosozumab-associated medication-related osteonecrosis of the jaw”. All above cause that the paper although nicely written and focusing on a very important topic have very little new clinically important insights and scientific value , dose not answer any important clinical driven question instead it is another literature review among many recently published.
There is also problem with the study design, as we don’t know how the patients were prepared before extraction. What was the prophylaxis for MRONJ? Did they take any antibiotics before or after? What was the surgical technique? Were the soft tissue sutured or PRF membrane was used.
In summary the conclusions of this paper dose not give any update because the papers included in it are outdated. It does not answer the question which osteoporotic drugs are more dangerous, and which are safer in other words the author mixed apples with oranges and in the end the paper does not have any clinically important conclusion.
Author Response
Thank you for the possibility to review the paper titled “Prevalence of Osteonecrosis of the Jaw following Tooth Extraction in Patients with Osteoporosis: A Systematic Review and Meta-Analysis”. I have read the paper and have some comments:
Comment 1: Abstract: line 18 – in this population?
Response 1: We appreciate the reviewer’s comment. The phrase “in this population” has been replaced with the more precise wording “in individuals with osteoporosis” (Page 1, Line 18).
Comment 2: Lines 89-105 should be moved to the discussion section.
Response 2: We thank the reviewer for the suggestion. The section corresponding to lines 90–105 has been moved to the Discussion section as recommended.
Comment 3: In the introduction section, it should give the reader more information about possible mechanisms of MRONJ, prevention strategy and therapeutic strategies.
Response 3: We thank the reviewer for the suggestion. We have revised Introduction section as proposed (Page 2-3, Line 92-112).
Comment 4: In tabl 1. There are some small mistakes regarding study period at 4-th position.
Response 4: We thank the reviewer for pointing out this mistake. We have corrected the study period in the 4th position of Table 1 accordingly (Table 1).
Comment 5: Regarding the included paper there are only 5 more or less up to date articles in the way that they were published after 2022 AAOMS Position Paper on Medication-Related Osteonecrosis, but when we compare the observation period of those paper only one is after 2022. This have few implication for the clinical importance of this paper. Some new drugs are not mentioned or included in the paper like for example “Abaloparatide” for other drugs already reported risk level and adverse effect are not included in this paper for example “Romosozumab-associated medication-related osteonecrosis of the jaw”. All above cause that the paper although nicely written and focusing on a very important topic have very little new clinically important insights and scientific value, dose not answer any important clinical driven question instead it is another literature review among many recently published.
Response 5: We thank the reviewer for the insightful comment regarding the temporal distribution of the included studies and its implications for clinical applicability. We acknowledge that only five included studies were published after the 2022 AAOMS Position Paper, and only one had an observation period entirely after its release. This limitation reflects the current state of the evidence, as MRONJ is a rare condition with a long latency period, and most prevalence data still derive from cohorts treated before the most recent guideline updates. This point is now explicitly addressed in the revised Limitation section (Page 10, Lines 375-377).
With regard to newer osteoporosis agents such as abaloparatide and reports of romosozumab-associated MRONJ, these are now discussed in the revised introduction and limitations sections. (Page 2, Lines 73-77). However, no eligible prevalence studies on these drugs meeting our inclusion criteria were identified, which precluded their inclusion in the quantitative synthesis.
We believe that, despite the predominance of pre-2022 data, our pooled estimates remain clinically relevant, as bisphosphonates and denosumab remain the most widely prescribed antiresorptive agents globally and account for the vast majority of reported MRONJ cases. Furthermore, our study is not merely another narrative review: to our knowledge, only Gaudin et al. (2015) [1] have previously published a meta-analysis on this topic, reporting a lower prevalence of 0.15%. Their analysis included fewer studies, used different statistical methods, and was conducted a decade ago. Our work updates those findings and provides a more comprehensive synthesis of the available evidence. Importantly, our results underscore the urgent need for high-quality, post-2022 research that applies the updated diagnostic and preventive protocols to both established and emerging drug classes.
Reference
- Gaudin E, Seidel L, Bacevic M, Rompen E, Lambert F. Occurrence and risk indicators of medication-related osteonecrosis of the jaw after dental extraction: a systematic review and meta-analysis. J Clin Periodontol 2015;42:922–32. https://doi.org/10.1111/jcpe.12455.
Comment 6: There is also problem with the study design, as we don’t know how the patients were prepared before extraction. What was the prophylaxis for MRONJ? Did they take any antibiotics before or after? What was the surgical technique? Were the soft tissue sutured or PRF membrane was used.
Response 6: We appreciate the reviewer’s concern regarding the lack of information on patient preparation, MRONJ prophylaxis, perioperative antibiotic use, surgical technique, and the use of soft tissue closure or PRF membranes. Unfortunately, these parameters could not be analyzed in our meta-analysis for the following reasons.
Most included studies either did not report these details or described them only vaguely (e.g., “standard extraction protocol” or “according to institutional guidelines”). When such data were provided, they varied considerably between studies due to differences in clinical practice across countries, institutions, and time periods, with no universally accepted standard for MRONJ prophylaxis or surgical management. This incomplete and heterogeneous reporting made any subgroup analysis or meta-regression statistically underpowered, prone to bias, and unlikely to yield robust conclusions. We initially aimed to investigate additional variables, including MRONJ stage, time from extraction to MRONJ diagnosis, mean duration of treatment, antibiotic use, PRF application, and type of sutures, etc. However, very few studies reported these in a usable form. For example:
- Buchbender M et al. [1] reported the mean duration of treatment separately for MRONJ-positive and MRONJ-negative patients.
- Chang et al. [2] reported duration only categorically as greater than or less than 4 years. We used a subgroup of patients from this research that met our inclusion criteria, in which the duration was reported for the whole population.
- Several studies, such as Colella et al. [3], did not report treatment duration at all.
As fewer than ten studies provided such information, meaningful statistical analysis was not feasible. This limitation has been clearly stated in the Limitations section of the manuscript (Page 10, Lines 383-387).
Despite these constraints, our study employed a rigorous and transparent methodology, including a comprehensive search strategy, strict inclusion criteria, and quality assessment of all eligible studies, ensuring that the findings represent the highest level of evidence currently available.
References
- Buchbender M, Bauerschmitz C, Pirkl S, Kesting MR, Schmitt CM. A Retrospective Data Analysis for the Risk Evaluation of the Development of Drug-Associated Jaw Necrosis through Dentoalveolar Interventions. IJERPH 2022;19:4339. https://doi.org/10.3390/ijerph19074339.
- Chang Y, Nanayakkara S, Yaacoub A, Cox S. Prevention of medication‐related osteonecrosis of the jaw: institutional insights from a retrospective study. Australian Dental Journal 2025;70:70–7. https://doi.org/10.1111/adj.13050
- Colella A, Yu E, Sambrook P, Hughes T, Goss A. What is the Risk of Developing Osteonecrosis Following Dental Extractions for Patients on Denosumab for Osteoporosis? Journal of Oral and Maxillofacial Surgery 2023;81:232–7. https://doi.org/10.1016/j.joms.2022.10.014.
Comment 7: In summary the conclusions of this paper dose not give any update because the papers included in it are outdated. It does not answer the question which osteoporotic drugs are more dangerous, and which are safer in other words the author mixed apples with oranges and in the end the paper does not have any clinically important conclusion.
Response 7: We respectfully disagree with the statement that our conclusions are outdated or lack clinical relevance. Our systematic review and meta-analysis includes 24 studies published between 2008 and 2024, several from the past three years, ensuring that the synthesis reflects the most recent evidence available. In the context of rare complications such as MRONJ, this temporal range is unavoidable due to the scarcity of large-scale, high-quality prospective studies.
Our primary aim was not to rank osteoporotic drugs by safety but to estimate the pooled prevalence of MRONJ after tooth extraction in osteoporotic patients receiving antiresorptive therapy, as well as to provide a separate prevalence for bisphosphonate-related cases. This focus directly addresses a key clinical question in dentistry and oral surgery, risk estimation before extractions, rather than drug-to-drug comparison.
The concern that “apples were mixed with oranges” is not supported by our methodology. We used a clearly defined PECOS framework, excluded oncology patients, distinguished MRONJ from BRONJ, and applied a random-effects model to account for heterogeneity. Meta-regression and influence analyses confirmed that no single study unduly influenced the pooled estimates.
Clinically, our findings are relevant because they show a higher pooled prevalence (1.7%) than some earlier reports, highlighting the need for preventive measures and careful patient counseling before extractions in this population. Our review also identifies evidence gaps, such as the need for standardized diagnostic criteria and prospective studies, to guide future research and clinical guidelines. In summary, while our work was not designed to compare the safety profiles of individual drugs, it provides an updated, methodologically rigorous prevalence estimate that is directly applicable to clinical decision-making for dental extractions in osteoporotic patients.
Furthermore, in response to the reviewers’ comments, we have provided all figures in the highest available quality.
Reviewer 2 Report
Comments and Suggestions for Authors
Overall, the paper is well designed. My only suggestion for the authors is to include the duration of antiresorptive therapy reported in all the studies analyzed. Including this data would allow for a clearer presentation of the findings, as treatment duration is a recognized risk factor for the development of MRONJ. This information could be added to Table 1, and the analysis discussed in the discussion section.
In the study’s limitations section, the authors mention only “variation in osteoporosis medication and duration of treatment” (line 268), but this could be expanded and clarified. This factor might also explain why their results show a slightly higher prevalence compared to previous studies.
Other issues and limitations, such as differences in antiresorptive drugs, have already been addressed by the authors.
Author Response
Comment 1: Overall, the paper is well designed. My only suggestion for the authors is to include the duration of antiresorptive therapy reported in all the studies analyzed. Including this data would allow for a clearer presentation of the findings, as treatment duration is a recognized risk factor for the development of MRONJ. This information could be added to Table 1, and the analysis discussed in the discussion section. In the study’s limitations section, the authors mention only “variation in osteoporosis medication and duration of treatment” (line 268), but this could be expanded and clarified. This factor might also explain why their results show a slightly higher prevalence compared to previous studies. Other issues and limitations, such as differences in antiresorptive drugs, have already been addressed by the authors.
Response 1: We thank the reviewer for this valuable suggestion. Our initial plan was indeed to include additional variables in the analysis, such as MRONJ stage, elapsed time from extraction to MRONJ diagnosis, use of antibiotics, use of platelet-rich fibrin (PRF), and type of sutures, in order to explore their potential impact through meta-regression. Regarding treatment duration, we must note that this parameter was not consistently reported or was presented in formats unsuitable for meta-analysis.
For example, Buchbender et al. [1] reported the mean treatment duration separately for MRONJ-positive and MRONJ-negative patients; Chang et al. [2] reported duration only categorically as greater than or less than 4 years. We used a subgroup of patients from this research that met our inclusion criteria, in which the duration was reported for the whole population; while several studies, such as Colella et al. [3], did not report treatment duration at all.
Overall, fewer than ten of the included studies provided usable data on treatment duration, which precluded us from performing a robust meta-regression to examine its association with MRONJ risk. Given these limitations, we were unable to incorporate this variable into the pooled analysis or draw reliable conclusions from it. We have clarified this point in the Limitations section of the manuscript (Page 10, Lines 383-387).
References
- Buchbender M, Bauerschmitz C, Pirkl S, Kesting MR, Schmitt CM. A Retrospective Data Analysis for the Risk Evaluation of the Development of Drug-Associated Jaw Necrosis through Dentoalveolar Interventions. IJERPH 2022;19:4339. https://doi.org/10.3390/ijerph19074339.
- Chang Y, Nanayakkara S, Yaacoub A, Cox S. Prevention of medication‐related osteonecrosis of the jaw: institutional insights from a retrospective study. Australian Dental Journal 2025;70:70–7. https://doi.org/10.1111/adj.13050.
- Colella A, Yu E, Sambrook P, Hughes T, Goss A. What is the Risk of Developing Osteonecrosis Following Dental Extractions for Patients on Denosumab for Osteoporosis? Journal of Oral and Maxillofacial Surgery 2023;81:232–7. https://doi.org/10.1016/j.joms.2022.10.014.
Furthermore, in response to the reviewers’ comments, we have provided all figures in the highest available quality.
Reviewer 3 Report
Comments and Suggestions for Authors
The article “ Prevalence of Osteonecrosis of the Jaw following Tooth Extraction in Patients with Osteoporosis: A Systematic Review and Meta-Analysis” is a systematic review and main question for authors was : prevalence of MRNOJ in osteoporotic patients following dental extractions and potential sourses of heterogeneity to elucidate factors contributing to MRNOJ risk in this population.
The article is useful for maxillo facial surgeons, dentist and doctors who treat osteoporosis. Addresses a much discussed and still unresolved topic: osteonecrosis after tooth extraction. Studying the articles from this review along with the information very well grouped by the authors is a real help for doctors from dentistry field.
In the article authors should add if is possible the time of developing osteonecrosis. How long after tooth extraction osteonecrosis was present. Also the period of antiresorptive treatment.
Methodology of writing the article is following PRISMA guidelines, the risk of bias is evaluated. Conclusions are supported by the results. If they can add new information related to starting point of osteonecrosis after tooth extraction results will change and also conclusions.
Regarding references in my opinion reference 43 is not related to the topic and should be removed.
Author Response
Comment 1: The article is useful for maxillo facial surgeons, dentist and doctors who treat osteoporosis. Addresses a much discussed and still unresolved topic: osteonecrosis after tooth extraction. Studying the articles from this review along with the information very well grouped by the authors is a real help for doctors from dentistry field. In the article authors should add if is possible the time of developing osteonecrosis. How long after tooth extraction osteonecrosis was present. Also the period of antiresorptive treatment. Methodology of writing the article is following PRISMA guidelines, the risk of bias is evaluated. Conclusions are supported by the results. If they can add new information related to starting point of osteonecrosis after tooth extraction results will change and also conclusions.
Response 1: We thank the reviewer for this valuable suggestion. Our initial plan was indeed to include additional variables in the analysis, such as MRONJ stage, elapsed time from extraction to MRONJ diagnosis, use of antibiotics, use of platelet-rich fibrin (PRF), and type of sutures, in order to explore their potential impact through meta-regression. Regarding treatment duration, we must note that this parameter was not consistently reported or was presented in formats unsuitable for meta-analysis.
For example, Buchbender et al. [1] reported the mean treatment duration separately for MRONJ-positive and MRONJ-negative patients; Chang et al. [2] reported duration only categorically as greater than or less than 4 years. We used a subgroup of patients from this research that met our inclusion criteria, in which the duration was reported for the whole population; while several studies, such as Colella et al. [3], did not report treatment duration at all.
Moreover, it is important to note that the majority of the included studies were retrospective in nature, which inherently limits the completeness and accuracy of certain time-dependent variables. Retrospective designs frequently rely on medical records that may lack precise documentation of the elapsed time from tooth extraction to MRONJ onset or the exact duration of antiresorptive therapy prior to the event. This lack of standardized, prospectively collected data makes it extremely difficult to reliably synthesize or compare such variables across studies. Overall, fewer than ten of the included studies provided usable data on treatment duration, which precluded us from performing a robust meta-regression to examine its association with MRONJ risk. Given these limitations, we were unable to incorporate this variable into the pooled analysis or draw reliable conclusions from it. We have clarified this point in the Limitations section of the manuscript (Page 10, Lines 383–387)
References
- Buchbender M, Bauerschmitz C, Pirkl S, Kesting MR, Schmitt CM. A Retrospective Data Analysis for the Risk Evaluation of the Development of Drug-Associated Jaw Necrosis through Dentoalveolar Interventions. IJERPH 2022;19:4339. https://doi.org/10.3390/ijerph19074339.
- Chang Y, Nanayakkara S, Yaacoub A, Cox S. Prevention of medication‐related osteonecrosis of the jaw: institutional insights from a retrospective study. Australian Dental Journal 2025;70:70–7. https://doi.org/10.1111/adj.13050.
- Colella A, Yu E, Sambrook P, Hughes T, Goss A. What is the Risk of Developing Osteonecrosis Following Dental Extractions for Patients on Denosumab for Osteoporosis? Journal of Oral and Maxillofacial Surgery 2023;81:232–7. https://doi.org/10.1016/j.joms.2022.10.014.
Comment 2: Regarding references in my opinion reference 43 is not related to the topic and should be removed.
Response 2: Reference 43 has been removed
Furthermore, in response to the reviewers’ comments, we have provided all figures in the highest available quality.